# Comparative Genomics of *Pseudomonas aeruginosa* Strains Isolated from Different Ecological Niches

**DOI:** 10.3390/antibiotics12050866

**Published:** 2023-05-07

**Authors:** Jessica Gómez-Martínez, Rosa del Carmen Rocha-Gracia, Elena Bello-López, Miguel Angel Cevallos, Miguel Castañeda-Lucio, Yolanda Sáenz, Guadalupe Jiménez-Flores, Gerardo Cortés-Cortés, Alma López-García, Patricia Lozano-Zarain

**Affiliations:** 1Posgrado en Microbiología, Centro de Investigaciones de Ciencias Microbiológicas, Instituto de Ciencias, Benemérita Universidad Autónoma de Puebla, Puebla 72570, Mexico; jessica.gomezmar@alumno.buap.mx (J.G.-M.); rosa.rocha@correo.buap.mx (R.d.C.R.-G.); miguel.castaneda@correo.buap.mx (M.C.-L.); alma.lopez@correo.buap.mx (A.L.-G.); 2Programa de Genómica Evolutiva, Centro de Ciencias Genómicas, Universidad Nacional Autónoma de México, Cuernavaca 62210, Mexico; bellolop@ccg.unam.mx (E.B.-L.); mac@ccg.unam.mx (M.A.C.); 3Área de Microbiología Molecular, Centro de Investigación Biomédica de La Rioja (CIBIR), 26006 Logroño, Spain; ysaenz@riojasalud.es; 4Laboratorio Clínico, Área de Microbiología, Hospital Regional Instituto de Seguridad y Servicios Sociales de los Trabajadores del Estado, Puebla 72570, Mexico; lupitajf@live.com.mx; 5Department of Microbiology and Environmental Toxicology, University of California at Santa Cruz, Santa Cruz, CA 95064, USA; gccortes@ucsc.edu

**Keywords:** *Pseudomonas aeruginosa*, antimicrobial resistance, ecological niches, comparative genomic

## Abstract

The *Pseudomonas aeruginosa* genome can change to adapt to different ecological niches. We compared four genomes from a Mexican hospital and 59 genomes from GenBank from different niches, such as urine, sputum, and environmental. The ST analysis showed that high-risk STs (ST235, ST773, and ST27) were present in the genomes of the three niches from GenBank, and the STs of Mexican genomes (ST167, ST2731, and ST549) differed from the GenBank genomes. Phylogenetic analysis showed that the genomes were clustering according to their ST and not their niche. When analyzing the genomic content, we observed that environmental genomes had genes involved in adapting to the environment not found in the clinics and that their mechanisms of resistance were mutations in antibiotic resistance-related genes. In contrast, clinical genomes from GenBank had resistance genes, in mobile/mobilizable genetic elements in the chromosome, except for the Mexican genomes that carried them mostly in plasmids. This was related to the presence of CRISPR-Cas and anti-CRISPR; however, Mexican strains only had plasmids and CRISPR-Cas. *bla*_OXA-488_ (a variant of *bla*_OXA50_) with higher activity against carbapenems was more prevalent in sputum genomes. The virulome analysis showed that *exoS* was most prevalent in the genomes of urinary samples and *exoU* and *pldA* in sputum samples. This study provides evidence regarding the genetic variability among *P. aeruginosa* isolated from different niches.

## 1. Introduction

*Pseudomonas aeruginosa* is a ubiquitous Gram-negative bacterium that can survive in different environments such as soil, food, and hospitals [1,2]. This opportunistic pathogen is the leading cause of nosocomial infections such as ventilator-associated pneumonia, catheter-associated urinary tract infections, bloodstream infections, and surgical site infections [3].

*P. aeruginosa* infections are difficult to treat because this bacterium has developed resistance to many antibiotics. As a result, it has been included as one of three bacterial species on the list of “priority pathogens” urgently requiring the development of new antibiotics by the World Health Organization [4]. Many factors are responsible for the multidrug resistance (MDR) of *P. aeruginosa*, such as low outer-membrane permeability, production of beta-lactamases, chromosomal DNA mutations, and acquisition of resistance genes through mobile genetic elements [5,6].

The pathophysiology of the infections caused by *P. aeruginosa* involves several mechanisms, including adhesion, invasion, evasion of the immune response, and antibiotic resistance, making these infections difficult to treat [7]. Furthermore, the persistence of *P. aeruginosa* in infections is also related to its MDR as well as its virulence. The latter is exhibited as a result of the production of a wide variety of virulence factors, including, lipopolysaccharide, flagella, type IV pili, elastase (LasA, LasB), various exotoxins (ExoY, ExoS, ExoT, ExoU), alkaline protease, phospholipases, small molecules (phenazines, rhamnolipids and cyanide) and siderophores (pyoverdine and pyochelin), which are regulated by cell-to-cell signaling systems [8]. Some studies have reported that the genes associated with virulence are highly conserved among *P. aeruginosa* strains; however, variations in the presence of *exoU* and *exoS* genes (cytotoxins secreted by the type III secretion system) have been reported in some strains because these are acquired by horizontal transfer [9,10].

The genome size of *P. aeruginosa* ranges from 5.5 to 7 Mbp, which is relatively large compared to other bacterial genera [11]. Additionally, *P. aeruginosa* exhibits a mosaic structure composed of conserved genes in all strains, regardless of their origin (clinical or environmental), and variable accessory genes between strains, which are involved in adaptation to different ecological niches. Other authors have suggested that the content of the accessory genome determines environmental adaptability [11,12,13]. However, most studies have focused on strains isolated from eye and cystic fibrosis infections [14,15]. The authors report that these studies were conducted because *P. aeruginosa* plays an important role in cystic fibrosis infections, whereby, these types of strains have been extensively sequenced [14,15].

Mathee et al. compared the genomes of two strains of *P. aeruginosa* isolated from cystic fibrosis patients and three reference strains, establishing evidence that *P. aeruginosa* modifies its genomic composition to allow expansion and adaptability to different niches [14]. Another study comparing the genomes of *P. aeruginosa* strains isolated from ocular infection with reference strains suggested that the acquisition of resistance and virulence genes (*exoU*) through horizontal gene transfer (HGT) could play an important role in adaptation and pathogenesis in humans [16]. In 2018, Subedi, proposed that genomic diversity not only exists between strains isolated from different anatomical sites (eye and cystic fibrosis) but also between strains isolated from different geographical locations [15]. On the other hand, Dettman and Kassen (2020) studied strains from four unique ecological niches (environment, animal and human infections, and infections from patients with cystic fibrosis). They proposed that strains recovered from any ecological niche can adapt and develop to cause an infection in patients with cystic fibrosis [17].

In this study, we compare genomes of four strains isolated from urine and sputum samples from a hospital in Mexico with genomes from the GenBank database of three sources of isolation (urinary tract, sputum, and environmental) to identify resistance genes, mobile genetic elements, and virulence genes and to explore relationships among them despite their unique niches.

## 2. Results

### 2.1. General Features of the Genomes

The results of the de novo assemblies of strains PE21, PE52, PE63, and PE83 from the Mexican Hospital are shown in Appendix A. We found an average G+C content of 65.9% and genome sizes ranging from 6,235,658 to 7,388,095 bp (Appendix A). In contrast, the average genome sizes of urinary, sputum, and environmental strains (from different countries) from the GenBank database were 6,831,303 bp, 7,013,391 bp, and 6,627,270 bp, respectively (Appendix A). Furthermore, it is important to highlight that the genome size of the PE21 strain is larger than the genomes in GenBank.

### 2.2. Multilocus Sequence Typing (MLST)

The strains isolated from a Mexican hospital had the following STs: PE21 and PE83 belonged to ST167, while PE52 and PE63 belonged to ST2731 and ST549, respectively (Appendix A). Therefore, we investigated the clonal relationship between these STs and high-risk STs, and none of these were high-risk.

We observed different ST distributions between clinical (urinary and sputum) and environmental genomes. High-risk ST235 and ST773 (3 and 3 genomes, respectively) were found in urine-derived genomes. In sputum-derived genomes, we found ST235 and ST357(4 and 3, respectively). On the other hand, ST27 (high-risk clone) and ST316 were found in environmental genomes (3 genomes each). It is important to highlight that ST235 and ST773 were exclusively found in clinical genomes, while ST27 was found only in environmental genomes (Appendix A). Lastly, one environmental genome (L10) belonged to ST253, which is the same ST as the reference strain and highly virulent PA14.

### 2.3. Genomic Comparison of Strains Recovered from Mexican Hospital

Pangenome analysis with Roary of PE21 isolated from sputum and PE52, PE63, and PE83 isolated from urine identified a total of 7793 genes, of which 5327 genes belong to the core, 0 to the soft core, 2466 to shell genes, and 0 to cloud genes, based on the classification given by Collins and Higgs, 2012 [18]. We also identified the number of shared and unique genes among the genomes of the strains using a Venn diagram (Figure 1). The results showed that strain PE83 shared 1068 genes with strain PE21, while strain PE63 shared 141 genes with the reference strain PAO1. In contrast, PE21, PE52, and PE83 shared one, four, and zero genes, respectively, with the strain PAO1 (Figure 1).

On the other hand, the PE52 strain carried the most significant number of unique genes (609 genes), including some genes that shape the conjugation machinery (*traG*, *viB11*, *virB4*, *virB10*, *mobA*), resistance to quaternary ammonium compounds genes (*sugE*), nickel and cobalt resistance genes (*cnrA*), bleomycin resistance genes (*ble*), toxin and antitoxin genes (*colM*, *parD1*, *yhaV*), genes required for vibriobactin utilization (*viuB*), ATP-dependent DNA helicase gene (*pcrA*), and a large number of hypothetical proteins (*n* = 491 (80.6%)).

### 2.4. Analysis of the Pan-Genome of the Strains from the Mexican Hospital and of the GenBank

The pangenome analysis identified a total of 20,911 pan-genes, of which 4246 were core genes (genes present in 99–100% of the genomes), 898 were soft core genes (genes present in 95–99% of the genomes), 1709 were shell genes (genes present in 15–95% of the genomes), and 14,058 were cloud genes (genes present in 0–15% of the genomes) (Figure 2).

The dendrogram based on the accessory genome shows the clustering of the genomes belonging to the same ST but from different niches (Figure 2 and Appendix A). Two exceptions were found: one genome belonging to ST111 (F5677) was clustered in a clade with different ST, and the genomes with ST234 (97 and AJD2) were not clustered in the same clade (Figure 2 and Appendix A). This was also found in the phylogenetic trees based on the core genome, SNPs, and MLST (Appendix A).

The genome of PPF-1 (ST unknown) was clustered with ST316 genomes (Appendix A), and we found that the ST of PPF-1 varied only in one allele compared to ST316 (aroE_13 and aroE_8, respectively).

It is important to highlight that our analysis revealed that the genomes of environmental strains (HS9, DN1, JB2, JT86, N17-1, and SJTD-1) had genes in the soft core that some clinical strains do not have. These genes could be related to the survival of *P. aeruginosa* in the environment, such as genes of p-hydroxybenzoic acid efflux pump, methyl-accepting chemotaxis protein (*pctB*), serine/threonine protein kinase (*rdoA*), nitrite reductase, alkanesulfonate monooxygenase, multidrug/solvent efflux pump outer membrane protein (*mepC*), lactose transport system permease protein (*lacF*), glycogen synthase.

### 2.5. Antibiotic Resistance Genes

We screened the genomes of the strains to detect antibiotic-resistance genes. The genes *bla*_OXA-50_ and *bla*_PAO_, which confer beta-lactam resistance, *fosA* to fosfomycin, and *aph*(*3’*)*-lib* to aminoglycoside, were identified in all genomes studied, including the reference strains PAO1, PA14 (Figure 3). However, the *catB7* gene that confers resistance to chloramphenicol was found in all genomes except for the genomes B-I-1 and 60,503 (Figure 3).

Because *bla*_OXA-50_ was present in all genomes, an analysis of the variants of this family was carried out. We found that the most common variants in the urinary and environmental strains were *bla*_OXA-50_ and *bla*_OXA-494,_ respectively, and *bla*_OXA-488_ was the most common in sputum genomes (Appendix A).

On the other hand, we found genes encoding antibiotic resistance only in urinary genomes, such as *bla*_SPM-1_, *bla*_CARB-2_, and *bla*_CTX-M-30_, which confer beta-lactam resistance, and *arr2* and *arr7*, which confer rifampicin resistance. Additionally, we found variants of *aadA2* and *rmt* (*rmtB4*, *rmtB4*, *rmtB4*, *rmtD1*) that confer aminoglycoside resistance (Figure 3). Furthermore, we identified unique genes in sputum genomes, such as *bla*_KPC-2_, *bla*_GES_, *bla*_PME_, and *bla*_PER_, which confers beta-lactam resistance, *catB3*, which confer chloramphenicol resistance; *arr3* (rifampicin resistance); *msrE* (erythromycin resistance); *qnrVC6*, *qnrS2* (quinolone resistance); *aac*(*2’*)*-lia* (aminoglycoside resistance); and *tet*(*A*) (efflux pump conferring resistance to tetracyclines) (Figure 3).

*qnrVC1*, *tet*(G), *cmlA*, and variants of the *bla*_IMP_ and *bla*_OXA_ genes were found to be more prevalent in urinary genomes compared to sputum genomes. In contrast, *ant2*, *aadA1*, *crpP*, *dfrB*, *dfrA*, *floR*, *cmx*, and variants of the *bla*_NDM_ and *bla*_VEB_ were predominant in sputum genomes. *crpP* was the only acquired resistance gene found in environmental genomes (Figure 3).

All resistance genes found in the genomes of strains isolated from the Mexican hospital are shown in Appendix A. However, it is important to note that PE21 and PE83 were the only genomes carrying the *catA1* gene (chloramphenicol resistance) (Figure 3).

### 2.6. Gene Mutations Associated with Drug Resistance

Chromosomal antibiotic resistance mutations were searched in 81 genes. The analysis showed nine genes with antibiotic resistance-related mutations. All genomes showed polymorphisms in the *ampC* gene. Seven urinary genomes, three from sputum, and five from environmental showed mutations in the *ampD* gene. In addition, one urinary genome had a mutation in the *ampR* regulator gene, which is involved in the overproduction of the chromosomal AmpC β-lactamase (Figure 4).

The *oprD* gene and regulators that reduce OprD expression had mutations and premature stop codons in eleven urinary genomes, fifteen from sputum, and two environmental genomes. In addition, six genomes from sputum, seven from urine, and twelve from environmental samples had a loop L7-short that causes increased susceptibility to meropenem (Figure 4).

Mutations in DNA gyrase and topoisomerase IV genes (*gyrA*, *gyrB*, *parC*, and *parE*) associated with reduced susceptibility to quinolones were found only in urinary and sputum genomes (Figure 4).

The analysis of mutations in the four efflux pumps’ regulators showed only mutations with unknown effects on antibiotic resistance. Finally, mutations in the *parS* gene were found in only one urinary and three sputum genomes (Figure 4).

In the genomes of strains PE21, PE52, and PE83 were identified antibiotic resistance-related mutations in the *oprD* and *gyrA* genes. In addition, PE52 showed resistance-related mutations in the *gyrB* gene and PE21 and PE83 genomes in the *parC* gene (Figure 4).

### 2.7. Mobilome

The accessory genome comprises mobile genetic elements that are associated with bacterial environmental adaptation. We analyzed all genomes for the presence of insertion sequences (IS), simple transposons (Tn), composite transposons (CTn), integrative conjugative elements (ICEs), integrative mobilizable elements (IMEs), and plasmids. We found that urinary genomes had a higher number of IS, IMEs, and plasmids (216, 11, and 5), while sputum genomes had a higher number of Tn, CTn, and ICEs (13, 49, and 56, respectively). The environmental genomes showed fewer mobile genetic elements (Figure 5).

The genes *bla*_IMP_, *bla*_DIM_, *bla*_OXA_, *bla*_GES_, and *bla*_VIM_ were associated with classical and partial class 1 integrons, while genes *bla*_KPC_ and *bla*_NDM_, *bla*_SPM_ were associated with transposons. Furthermore, transposons carrying the *bla*_KPC-2_ were found in two plasmids from two genomes from sputum (R31 and SE5416) (Appendix A). The transposons and integrons carrying *bla*_OXA-101_, *bla*_CTX-M-30_, and *bla*_TEM-1b_ were found in a plasmid from a urinary genome (PB353), and *bla*_DIM-1_ was found in a plasmid from sputum (60503) (Appendix A). On the contrary, transposons or integrons carrying *bla*_GES-15_, *bla*_SPM-1_, *bla*_GES-1_, *bla*_PME-1_, *bla*_OXA-1_, *bla*_OXA-50_, *bla*_NDM-1_, *bla*_CARB-2_, *bla*_OXA-35_, *bla*_IMP-1_, *bla*_VIM-4_, *bla*_OXA-56_, and *bla*_IMP-13_ were found in integrative conjugative elements (ICEs) with a complete type 4 secretion system (T4SS) in urinary and sputum genomes (Appendix A). Lastly, the *catB7* gene was found in ICEs with T4SS from urinary and sputum genomes, while *crpP* was carried by ICEs in all genomes. In the Mexican hospital genomes, the *bla*_IMP_ gene was carried on plasmids.

### 2.8. Correlation between the Presence of Plasmids, Relaxases MOB, CRISPR-Cas, and Anti-CRISPR Systems

MOB_P_ and MOB_H_ were found in clinical (urine and sputum) and environmental strain genomes. On the other hand, MOB_Q_ was only found in the clinical genomes and MOB_C_ in one urinary genome (Appendix A). In the Mexican hospital genomes, we found MOB_H2_ in all genomes, MOB_P11_ in PE21, PE52, and PE83 genomes, and MOB_P14_ in PE63 (Appendix A).

CRISPR-Cas systems were found in 18 urinary, nine sputum, and eight environmental genomes, including genomes from the Mexican Hospital. CRISPR-Cas systems and plasmids were present in seven genomes (PE21, PE52, PE63, PE83, IMP-13, 60503, and SE5416) (Appendix A). On the other hand, anti-CRISPR, CRISPR-Cas systems, and plasmids were identified in only three genomes (one from urinary and two from sputum) (Appendix A).

The search for CRISPR-Cas systems in plasmids revealed their presence in the plasmid pPYO_TB (from the IMP-13 urinary genome) and the plasmid unnamed1 (from the DN1 environmental genome) (Appendix A). However, no such systems were found in the genomes of PE21, PE52, PE63, and PE83.

### 2.9. Virulence Genes

We analyzed 116 virulence genes associated with alginate biosynthesis and regulation, rhamnolipid biosynthesis, iron uptake, quorum sensing, proteases, and toxins using the Virulence Factor Database (VFDB). However, we included only thirteen virulence factors associated with urinary tract and lung infections in the heatmap. In addition, we used the virulence genes of the PAO1 strain as a reference, and the *exoU* gene of the PA14 strain was taken as a reference.

The genes for effector proteins of the type 3 secretion system analyzed were *exoS*, *exoT*, *exoU*, and *exoY*. Our results showed that the *exoS* gene was more prevalent in urinary genomes, whereas *exoU* was found mainly in sputum genomes, and *exoY* was predominant in both urinary and environmental genomes. Finally, the *exoT* gene was present in all genomes. The *exoS* y *exoU* genes are almost always mutually exclusive; however, in this study, both were present in sputum and an environmental genome (R31 and JT86). The *pvdE* gene (involved in pyoverdine synthesis), the exotoxin A gene (*toxA*), the elastase A gene (*lasA*), and the elastase B gene (*lasB*) were found in all genomes. The *aprA* gene, encoding an alkaline protease, was present in all genomes except PE52 and JB2 (urinary and environmental, respectively). The phospholipase genes *plcB* and *plcH* were found in all genomes, and the *plcN* gene was absent in only one urinary strain. On the contrary, the *pldA* gene was more frequently observed in sputum (Figure 6).

## 3. Discussion

*P. aeruginosa* is a versatile opportunistic pathogen capable of adapting to different ecological niches due to its variable arsenal of virulence factors and antibiotic resistance determinants [19], but also to the acquisition of genes by horizontal transfer. In fact, Kung and collaborators suggest that the content of the accessory genome determines environmental adaptability [12]. In this study, we compared the genomes of strains from a Mexican hospital with GenBank genomes of urine, sputum, and environmental isolates from different cities. First, we wanted to know the distribution of STs among the different niches. The ST is determined by combining the allelic variation of 7 *P. aeruginosa* housekeeping genes (*acsA*, *aroE*, *guaA*, *mutL*, *nuoD*, *ppsA*, and *trpE*) [20]. The high-risk clones are known to cause outbreaks of nosocomial infections worldwide, which are associated with poor clinical outcomes. This is due to their high levels of antibiotic resistance, pathogenicity, and virulence, as well as their enhanced ability to colonize and persist in a host [21]. In our analysis, we observed a high occurrence of high-risk ST235 and ST773 [22] in urinary and sputum strains, consistent with other studies in clinical strains recovered from various types of infections, including urinary and respiratory tract [23,24,25].

On the other hand, the high-risk ST27 [22] in this study was only found in environmental strains, which has also been reported in other studies and the MLST database [26,27,28], and in strains from humans and animals with lower frequencies [26,29,30].

The strains isolated from the Mexican hospital had different STs than strains from GenBank (ST167, ST2731, and ST549). eBURST analysis showed that ST2731 and ST549 of strains PE52 and PE63, respectively, did not have a relationship with the international high-risk clones. On the other hand, ST167 (strains PE21 and PE83) is derived from group 0, where high-risk ST111 is also found; however, they are not within the same clonal complex because they only shared two alleles [31].

To know the worldwide dissemination of the ST167, we consulted the MLST database, founding four strains, of which two were clinical strains from Mexico (Accession numbers: GCF_000795625.1and GCF_000794515.1) carrying *bla*_IMP-15_ and *bla*_IMP-62_; one strain from the United States (Accession number: GCF_000480475.1) and one strain from a country no reported (Accession number: GCA_021693455.1) (Appendix A). Based on the criteria for defining a high-risk clone, ST167 could be considered as a “local high-risk clone”, because while it has not been reported causing infections worldwide, it is associated with nosocomial infections in Mexico [32].

The pangenome represents the total number of genes in a study group [33] and based on the persistence of genes in the genome, it was divided into four classes: (1) core genome, (2) soft-core genome, (3) shell genome, and (4) cloud genome [18,34]. The pangenome of the sixty-five genomes analyzed consists of 20,911 genes. In other studies that included 17, 23, and 18 genomes, the pangenome size was smaller (9344, 9786, and 12,775 genes, respectively) [15,35,36], while the analysis of the pangenome with a size of 54,272 genes used 1311 genomes [37].

These differences may be influenced by the number of genomes analyzed in each study because the pangenome increases with the number and diversity of strains added to the analysis [15]. On the other hand, the core genome comprises genes involved in bacterial survival, and its size decreases concurrently with the addition of genomes to the analysis [33]. In our study, the core genome size comprised 4246 genes; in contrast, other studies have reported larger core genomes 5233 [35], 4910 [15], and 5109 [38] genes in strains isolated from different sources. Other factors that could influence this are the diverse nature of the strains and the different annotation tools used [15].

The soft-core, shell, and cloud genomes showed 898, 1709, and 14,058 genes, respectively, which was different compared with other studies that included strains isolated from various infection sites [39,40].

The accessory genome is composed of genes acquired through horizontal transfer due to exposure of the bacterium to its host, environment, or other bacteria, providing adaptative advantages to the bacterium [41]. Therefore, we constructed a dendrogram based on the accessory genome and did not observe any correlation with the isolation site; these genomes were clustered based on their ST. However, since the tree’s construction was based on the presence or absence of accessory genome genes, it could indicate that strains grouped in the same clade also carry similar acquired genes in their accessory genome. To corroborate this, we constructed phylogenetic trees based on the core genome, SNPs, and MLST, where we could observe that strains clustered again concerning their ST; this was also reported in another study [42].

Comparing the genome of the strains from the Mexican Hospital revealed that each strain had strain-specific genes, with strain PE52 holding the largest number of these genes. These strain-specific genes are typically involved in niche adaptation [43]. On the other hand, previous studies have reported that strains PE21 and PE83 exhibit similar phenotypic profiles of resistance and genotypic characteristics [44]. In the genome comparison, we observed that they also shared many genes.

We analyzed antimicrobial resistance and virulence genes because the accessory genome is often composed of these genes [45]. We found that the genes *bla*_OXA-50_ and *bla*_PAO_ (beta-lactam resistance), *fosA* (fosfomycin resistance), and *aph*(*3’*)-*lib* (aminoglycoside resistance) were present on the chromosome of all strains studied, including reference strains PAO1 and PA14; in other studies, these genes have also been identified in all strains included [15,46], suggesting that they are conserved genes in *P. aeruginosa*.

On the other hand, the *catB7* gene (responsible for chloramphenicol resistance) was absent in only two genomes. This gene has been found exclusively in the chromosome of *P. aeruginosa* strains but not in other bacteria [15,47]; however, in other studies, the absence of this gene has already been reported in some strains of *P. aeruginosa* [36,48].

OXA-50 is an intrinsic class D oxacillinase of *P. aeruginosa* that has a narrow-spectrum hydrolysis profile against antibiotics such as ampicillin, benzylpenicillin, cephaloridine, cephalothin, nitrocefin, piperacillin, and imipenem [49]. The OXA-50 family consists of 43 variants according to the Beta-lactamase Database (until August 1 2022) [50]. In this study, we searched for the distribution of these variants among the three genomes groups; *bla*_OXA-50_ and *bla*_OXA-494_ were the most prevalent in the genomes of urinary and environmental strains, respectively. However, *bla*_OXA-488_ was the most frequently found in the genomes of sputum strains. Compared to OXA-50, OXA-488 is three times more efficient against benzylpenicillin and twice more efficient hydrolyzing imipenem [51]. This could be due to the continuous selection pressure caused by the indiscriminate use of beta-lactams, which has led to the emergence of OXA-50 variants with an improved hydrolysis spectrum against imipenem.

*P. aeruginosa* is also capable of acquiring antibiotic-resistance genes. In this study, we did not observe any correlation between the presence of these acquired resistance genes and the site of isolation of the clinical strains. However, some clinical genomes exhibited antibiotic-resistance genes that are commonly found in genera other than *Pseudomonas* [52,53,54]. This may be due to the fact that urinary tract and pulmonary infections are often polymicrobial [55,56], and the close contact and interaction with other species may have facilitated genetic exchange between bacteria coexisting in the same niche. These results support the findings of Freschi and collaborators [37], who showed that horizontal gene transfer events are involved in the acquisition of antibiotic resistance genes. Although this study provides knowledge about the behavior and dynamics of *P. aeruginosa* in different niches, a more significant number of genomes from other niches could be included for a better understanding.

It is important to highlight that *bla*_IMP_ was more predominant in strains isolated from urine, especially in strains isolated from the Mexican Hospital. This notable difference could be related to selection pressure within the hospital and the consequent selection of carbapenem-resistant strains carrying this resistance gene. In fact, this hospital reported *bla*_IMP_ as the most prevalent carbapenem-resistance gene [44]. In addition, bacteria in the same niche could be influencing the acquisition of this resistance gene.

On the other hand, *crpP* was found in both clinical and environmental genomes. However, it was the only acquired resistance gene identified in environmental strains, which is consistent with another study [57]. In addition, this gene was previously associated with resistance to the antibiotic ciprofloxacin [58]. However, a recent study has concluded that CrpP is not responsible for ciprofloxacin resistance in *E. coli* [59].

Interestingly, some antibiotic-resistance genes were associated with mobile/mobilizable genetic elements. Such as *bla*_IMP_, *bla*_DIM_, *bla*_OXA_, *bla*_GES_, *bla*_VIM_ genes that were associated with classical and partial class 1 integrons, consistent with other studies [42,60,61]; while *bla*_KPC_ and *bla*_NDM_, *bla*_SPM_ were associated with insertion sequences and transposons, similar to reported by other authors [62,63,64]. Commonly, transposons and integrons are localized within plasmids and integrative conjugative elements [65], which coincides with our findings. Notably, the *crpP* gene was initially identified on a plasmid [58]; however, in this and another study, it was localized in ICEs [66]. Similarly, *catB7* was identified in ICEs, which could be related to the loss of this gene in some genomes. The identification of antibiotic-resistance genes carried by mobile/mobilizable genetic elements suggests an increased possibility of horizontal gene transfer within or between different isolates, species, and environments [67]. On the other hand, the low number of MGEs in the genomes of environmental strains compared to the genomes of clinical strains, could be related to the fact that they did not have horizontally acquired resistance genes since these are regularly carried by MGEs.

The number of plasmids in *P. aeruginosa* found in this study is lower than in other bacteria, such as Enterobacteriales [68]. However, the urinary strains carried more plasmids, highlighting the strains from the Mexican hospital. This result suggests that plasmid-mediated resistance could be the main mechanism of resistance in this hospital.

CRISPR-Cas systems have been found to play an important role in shaping the accessory genome [69] because they restrict horizontal gene transfer and bacteriophage infection [70]. To know whether CRISPR-Cas systems would be play a role in plasmid acquisition, we correlated their presence and absence with the presence of plasmids. Although most genomes presented CRISPR-Cas systems but not plasmids, some genomes carried both, which may be due to the presence of anti-CRISPR-Cas proteins whose function is to inhibit the activity of these systems [71] or because the spacers could be targeted to other mobile genetic elements. However, we did not analyze CRISPR spacer sequences. In contrast, the Mexican strains had plasmids and CRISPR-Cas systems but did not present anti-CRISPR.

On the other hand, CRISPR-Cas systems have been associated with small genome sizes and reduced abundance of ICEs [72]. In contrast, the genomes of environmental strains had fewer CRISPR-Cas systems, a smaller genome size, and a reduced number of ICEs in comparison with the genomes of sputum and urinary strains.

Additionally, two plasmids carried CRISPR-Cas, which seemed interesting to us, so we searched for the type of system it had, finding that it was system type IV. This type of CRISPR-Cas has only been found in plasmids and other mobile genetic elements, and it has been hypothesized that its main function is to eliminate competing plasmids [73,74]. The above has been studied in archaeal and bacterial plasmids (including the *Gammaproteobacteria*), finding that many spacers of type IV CRISPR-Cas systems carried by plasmids matched sequences from other plasmids [74]. In plasmids of *Klebsiella pneumoniae*, something similar was found [75]; however, this remains unclear in *P. aeruginosa*.

It is important to note that some genomes only had orphan CRISPR arrays (lacking Cas), which are usually considered vestigial. However, it has been found that some CRISPR of this type in *E. coli* could be functional [76]; in *P. aeruginosa*, this remains unclear.

*P. aeruginosa* can also acquire resistance through chromosomal mutations. In this study, we found antibiotic resistance-related mutations in *ampD*, *ampR*, *oprD*, *gyrA*, *gyrB*, *parC*, *parE*, and *parS* genes. OprD is an outer membrane protein involved in the diffusion of small peptides, basic amino acids, and carbapenems into the cell [77]. However, alterations in the structure/expression or loss of OprD cause decreases in susceptibility to carbapenems [78]. Analysis of *oprD* revealed that carbapenem resistance-related mutations were mostly found in the genomes of urine and sputum strains, as reported in other studies with clinical strains [44,79]. In contrast, deletions in the *oprD* sequence can generate Loop L7-short, which is associated with meropenem susceptibility phenotype [79,80]; interestingly, this *oprD* genotype was mostly found in the genomes of environmental strains.

The resistance to fluoroquinolones may be due to mutations in quinolone resistance determinant regions (QRDRs) in DNA gyrase (*gyrA* and *gyrB*) and topoisomerase IV (*parC* and *parE*) subunits [81], which were observed in the genomes of urine and sputum strains but not in the genomes of environmental strains. The absence of these mutations in environmental strains could suggest that they were not exposed to selection pressure since, although the environment may suffer antibiotic contamination because of human activity [82], they are not exposed to the same strong selection pressure for antibiotics as the hospital bacteria.

The production of the cephalosporinase AmpC is an intrinsic mechanism of resistance to beta-lactam antibiotics in *P. aeruginosa*. However, it can be potentiated by mutations in *ampR* (transcription factor) and *ampD* (cytosolic amidase) that cause *ampC* derepression [78]. We observed that mutations in AmpD, which lead to AmpC derepression, were most prevalent in genomes of urinary strains (*n* = 7), followed by environmental (*n* = 5), and to a lesser extent in genomes of sputum strains (*n* = 3). In contrast, only one genome from a urinary strain carried a mutation in the *ampR* gene leading to AmpC derepression. On the other hand, *ampC* polymorphisms caused by mutations increase its hydrolytic activity towards cephalosporins [83]. AmpC polymorphisms were observed in almost all studied genomes except for two (one sputum strain and one environmental strain). To date, 533 Pseudomonas derived cephalosporinase (PDC) variants have been described (until December 2022) [50]. However, this study did not focus on searching for variants, and we do not know if their distribution could be related to the isolation sites of the studied genomes.

Some mutations in ParS (the sensor kinase of the dual-component ParR/S system) are related to colistin resistance and were observed only in clinical strains, being more prevalent in sputum strain genomes (*n* = 3). It is important to mention that our study has some limitations, such as the fact that we do not know the phenotypic profile of antibiotic resistance and cannot relate it to its resistance genotype.

It is important to note that environmental strains only carried intrinsic resistome genes, and some mutations in these genes were identified, which may indicate that mutations in resistance-related genes are the main mechanisms in environmental strains and not the acquisition of genes through horizontal transfer.

*P. aeruginosa* uses a type 3 secretion system to release effector toxins (ExoS, ExoT, ExoU, and ExoY) directly into host cells, which aid colonization and immune evasion. In this study, variation in the presence and absence of virulence genes was most evident in these effector proteins. *exoS* encodes a cytotoxin with GTPase activating protein (GAP) activity and adenosine diphosphate ribosyl transferase (ADPRT) activity [84] and is mostly found in the genomes of urinary tract strains.

On the other hand, ExoU is a cytotoxin with phospholipase A2 activity that causes lysis and necroptosis in epithelial cells, macrophages, and neutrophils [85]. ExoU production has been shown to contribute to the developing of severe pneumonia in a mouse model [86] and poor outcomes in patients with ventilator-associated pneumonia [87]. *exoU* was primarily found in strains isolated from sputum; however, in other studies, it has also been found in eye isolates [15,88]. Similarly, it has been suggested that because the *exoU* gene is absent in PAO1 and is carried by genomic islands, it may have been acquired through horizontal transfer [10] to provide adaptive advantages in its ecological niche.

For reasons that are not entirely clear, *exoU* and *exoS* are mutually exclusive. However, strains carrying both cytotoxins have been found in this and other studies [89,90]. One possible explanation suggested is that they provide enhanced fitness in distinct ecological niches [91].

ExoY is an adenylyl cyclase whose action disrupts the actin cytoskeleton, inhibits bacterial uptake by host cells, and increases endothelial permeability; however, the significance of ExoY in infections remains unclear [92]. The *exoY* gene was mostly observed in genomes of both urinary and environmental strains. However, other studies have also observed it in strains isolated from the eye and cystic fibrosis [15]. ExoT is the most prevalent effector in genomes of clinical and environmental strains of *P. aeruginosa* [19,93]. This study found it in 100% of the genomes analyzed.

On the other hand, the prevalence of *plcN*, *plcH*, *lasB*, and *toxA* genes was 100% (except for *plcN*) in the three isolation sources, contrary to what was observed by Hassuna et al. (2020) in isolates from respiratory tract infections [90]. On the other hand, *plcB* and *plcH* were detected in 98.7% (153/155) and 96.1% (149/155) of strains recovered from nosocomial infections and community-acquired infections [94], indicating that they are highly conserved in the *P. aeruginosa* genome. The reason for the differences in the distribution of these genes in different studies remains unclear. Finally, *pldA* was mostly found in the genomes of sputum strains. In another study, it was also mostly found in isolates responsible for an acute lung infection and, to a lesser extent, in urinary tract infection isolates [95]. In addition, it has been suggested to play a role in chronic lung infection in rats [96].

It is important to note that although virulence genes were more prevalent in clinical strains, they were also present in lower amounts in genomes of environmental strains, as has been observed by another author [97]. This would reinforce the understanding that environmental strains can also cause infections [17] and be a potential risk factor for human health [98] and highlights the prevalence of virulence genes that could vary about the isolation site.

## 4. Materials and Methods

### 4.1. Bacterial Genomes

Four strains of *P. aeruginosa* multidrug-resistant and carbapenem resistant recovered from a Mexican Hospital, were included in this study: strains PE52 (Accession number: JARDUU000000000), PE63 (Accession number: JARDUW000000000), and PE83 (Accession number: JARDUX000000000) were isolated from urine, and strain PE21 (Accession number: JARDUV000000000) was isolated from sputum. The phenotype of sensibility and some molecular resistance mechanisms were previously described [44]. Genomic DNA was extracted using the Wizard^®^ Genomic DNA Purification Kit (Promega Corporation. Ma, USA), and whole genome sequencing was made by Illumina Miseq 2 × 150 bp, with 5 million paired end reads in the ‘SNPSaurus Genomics to Genotype” https://www.snpsaurus.com/ (accessed on 1 September 2019). The quality of the paired-end reads it was measured by FastQC version 3.9.0 [99]. SPAdes version 3.9.0 [100] was used to generate the de novo assemblies, and the quality was analyzed with QUAST (Quality Assessment Tool for Genome Assemblies) [101]. Finally, the genome sequences were annotated with Prokka version 1.12 [102].

In addition, 59 complete genomes from the GenBank database were downloaded in order of appearance available until June 2021, and the sources of isolation were obtained from the submitters’ information in GenBank. For the comparison, we included 19 genomes from strains isolated from urinary samples, 20 from sputum samples (non-cystic fibrosis patients), and 20 isolated from the environment. In addition, the genomes of the reference strain *P. aeruginosa* PAO1 (Accession number: NC_002516.2) and PA14 (Accession number: CP000438.1) were included.

### 4.2. MLST (Multilocus Sequence Typing)

The sequence type (ST) of 65 genomes was identified using the MLST tool from the Center for Genomic Epidemiology https://cge.food.dtu.dk/services/MLST/ (accessed on 1 February 2020) [103], which utilizes the MLST allele sequence and profile data from PubMLST.org. The ST is determined by combining the allelic variation of 7 *P. aeruginosa* housekeeping genes (*acsA*, *aroE*, *guaA*, *mutL*, *nuoD*, *ppsA*, and *trpE*). The clonal complex of sequence types (ST) was determined with PHYLOViZ [104] using goeBURST (a refinement of the eBURST algorithm) to generate a complete minimal spanning tree (MST).

### 4.3. Genome Comparison of Strains Isolated from the Mexican Hospital

The genomes of strains PE21 isolated from sputum, PE52, PE63, and PE83 isolated from urine, and the reference strain PAO1, were included in this analysis. Roary version 3.12.0 [105] was used to identify genes shared between strains and unique genes. The Roary file “gene_presence_absence.csv” was used to generate the Venn diagram, and the Venn diagram was performed with the online tool “calculate and draw custom Venn diagrams” [106].

### 4.4. Comparative Analysis of Strains from Three Isolation Sources

Sixty-five strain genomes were included in the pangenome analysis. Roary version 3.12.0 [105] was used to perform pangenome analysis, which uses the GFF3 files generated by Prokka to identify the core genome, soft core, shell genes, and cloud genes. Roary_plots.py script was used to visualize the pangenome. The phylogenetic tree based on single nucleotide polymorphisms (SNPs) was built using Parsnp v1.2 [107] with the “-c” flag enabled, including the previous 65 strain genomes and PA7 as the outliner strain. iTOOL v6.5 [108] was used to visualize the ‘newick’ files generated by Roary and parsnp. Seven concatenated sequences of the STs were obtained for the MLST database. MEGA version 11.0.10 was used to construct the MLST dendrogram using the UPGMA method.

### 4.5. Resistome Analysis

ResFinder version 4.1 was used to detect antibiotic resistance genes using default parameters. Mutations associated with antibiotic resistance leading to amino acid changes were identified using cluster OMEGA [109]. Amino acid changes were searched manually using *P. aeruginosa* PAO1 genes as a reference. Heatmaps and graphs were constructed using the Bioconductor and ggplot2 packages, respectively [110,111] from RStudio version 1.4.1106 [112].

### 4.6. Virulence Analysis

Virulence-associated genes were identified using the Virulence Factor of Pathogenic Bacteria database (VFDB) [113]. The *P. aeruginosa* PA14 strain was used as a reference because it is *exoU* positive, which is considered highly virulent.

### 4.7. Mobile Genetic Elements (MGEs)

Integrative Conjugative Elements (ICE) and Integrative and Mobilizable Elements (IMEs) were identified with the Web tool for ICE/IME detection of bacterial genomes: ICEfinder) [114]. Furthermore, mobile genetic elements such as simple transposons (Tn), composite transposons (CTn), and insertion sequences (IS) were analyzed with Mobile Element Finder version 1.0.3 [115]. Placnetw [116] was used to determine plasmid sequences, and MOBscan [117] was used to search the relaxase MOB families in the genomes. In addition, the online tool CrisprCasFinder [118] was used to identify “clustered regularly interspaced short palindromic repeats” and CRISPR-associated proteins (CRISPR-cas) on both chromosomes and plasmids of all genomes. Finally, AcrFinder was used to search anti-CRISPR systems [119].

## 5. Conclusions

In the present work, we compare the genomes of strains isolated from urine, lungs, and environment. When performing the phylogenetic analysis of genomes from a Mexican hospital and from the GenBank from different niches, we observed that the genomes were clustering according to their ST and not the niche. The high-risk STs (ST235, ST773, and ST27) were present in genomes from the three niches except for the Mexican genomes, and these last three (ST167, ST2731, and ST549) differed from the GenBank genomes. However, the analysis of the genomic content showed that environmental genomes had genes involved in adaptation to the environment that were not present in clinical genomes besides, these genomes had mutations in antibiotic resistance-related genes as the main resistance mechanism. In contrast, clinical genomes had resistance genes reported in *Pseudomonas* and *Enterobacteria*, and these genes were in mobile/mobilizable genetic elements in the chromosome, except for the Mexican genomes, which carried them mostly in plasmids. This was in concordance with the presence of CRISPR-Cas and anti-CRISPR, except in the Mexican genomes that only had plasmids and CRISPR-Cas. On the other hand, when analyzing the *bla*_OXA-50_ variants, we found *bla*_OXA-488_, which has higher activity against carbapenems, was more prevalent in sputum genomes. The virulome analysis showed *exoS* was most prevalent in the urinary genomes and *exoU* and *pldA* in sputum. This study evidences the variability of genetic content among *P. aeruginosa* isolates from different niches.

## Figures and Tables

**Figure 1 antibiotics-12-00866-f001:**
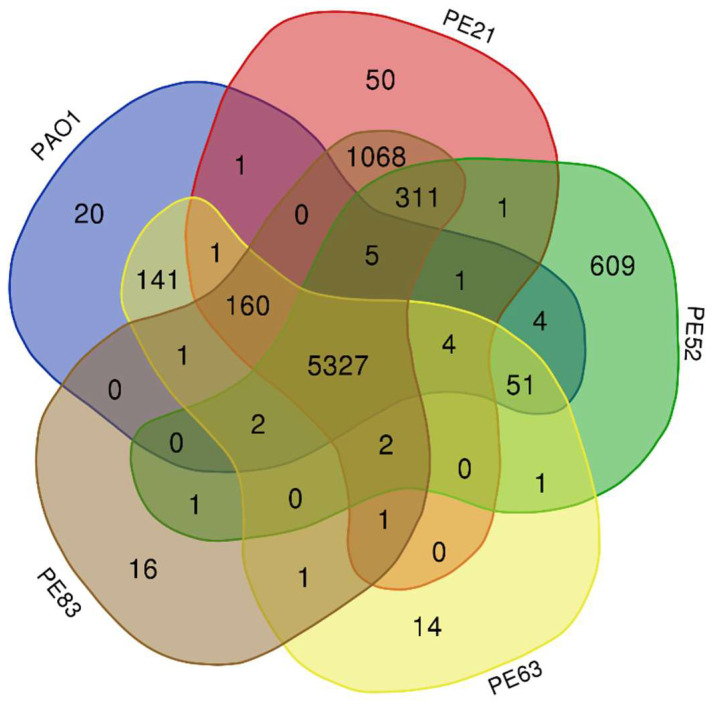
Genomic comparison of *P. aeruginosa* strains. The pan-genome analysis by Roary predicted the number of genes. The figure was created using Calculate and Draw Custom Venn Diagrams. The Venn diagram shows the number of unique and shared genes among the genomes of five strains: PAO1, PE21, PE52, PE52, PE63, and PE83 from the Mexican hospital.

**Figure 2 antibiotics-12-00866-f002:**
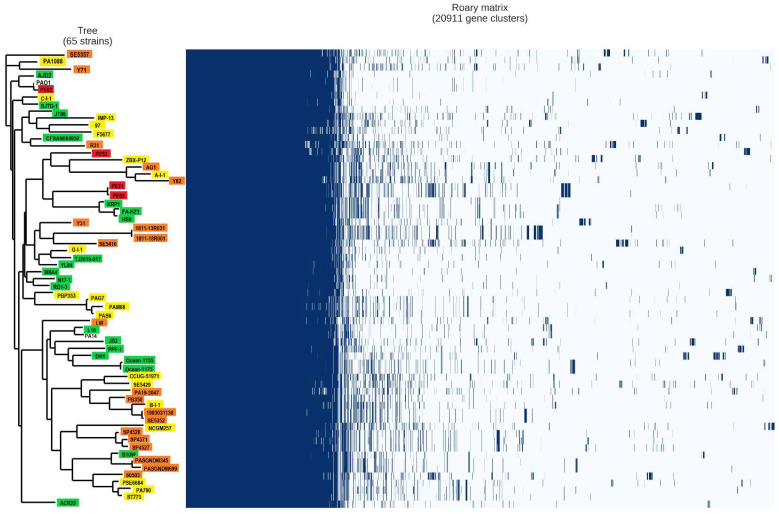
Graphical representation of the pangenome analysis of the 65 strains. The phylogenetic tree was constructed based on the accessory genomes of the sixty-five strains. Strain names were colored according to the isolation source. Yellow: urinary genomes; orange: sputum genomes; green: environmental genomes; red: Mexican genomes; reference genomes are in black.

**Figure 3 antibiotics-12-00866-f003:**
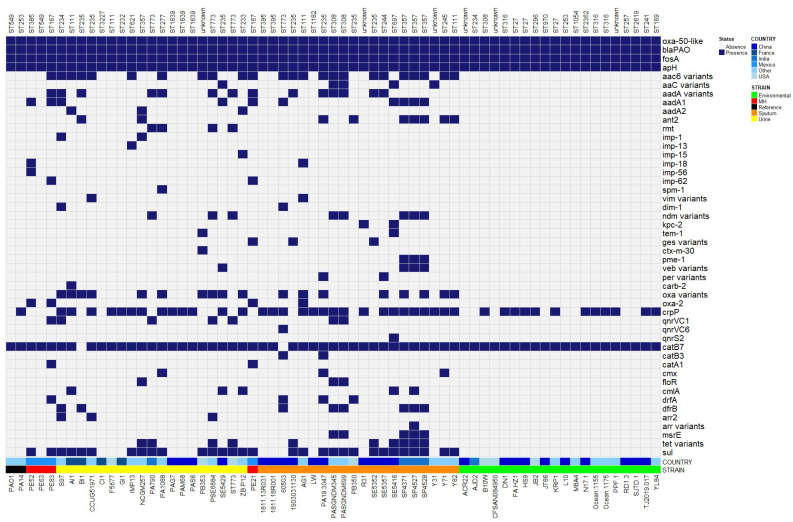
Presence and absence of resistance genes in urinary, sputum, and environmental genomes. Representation of resistance genes present and absent in the 65 genomes from urinary, sputum, and environmental. *P. aeruginosa* PAO1 and PA14 were used for comparison. ResFinder detected a total of 47 genes. The heatmap was constructed with the Bioconductor package in Rstudio. MH means Mexican Hospital.

**Figure 4 antibiotics-12-00866-f004:**
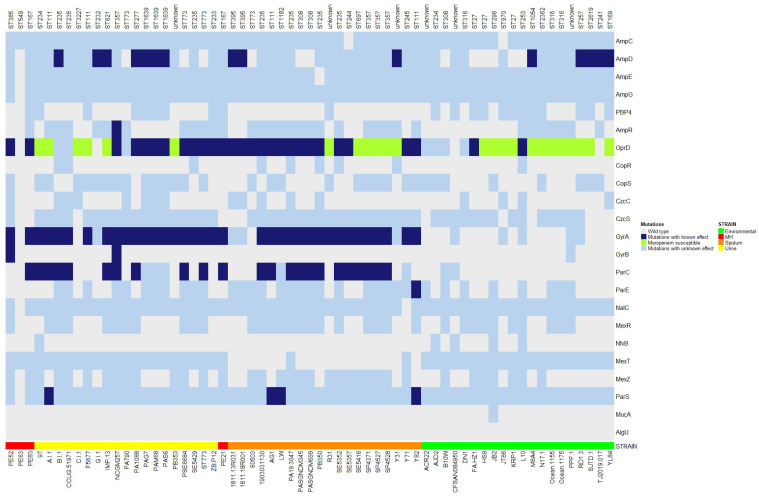
Mutational resistance of the 65 urinary, sputum, and environmental genomes. Graphical representation of mutations associated with antibiotic resistance (navy color), mutations with an unknown effect on antibiotic resistance (blue light), and mutations in *oprD* associated with susceptibility to meropenem (green) and wild-type genes (gray). *P. aeruginosa* PAO1 and PA14 were used as reference genomes. The heatmap was constructed with the Bioconductor package in Rstudio. MH means Mexican Hospital.

**Figure 5 antibiotics-12-00866-f005:**
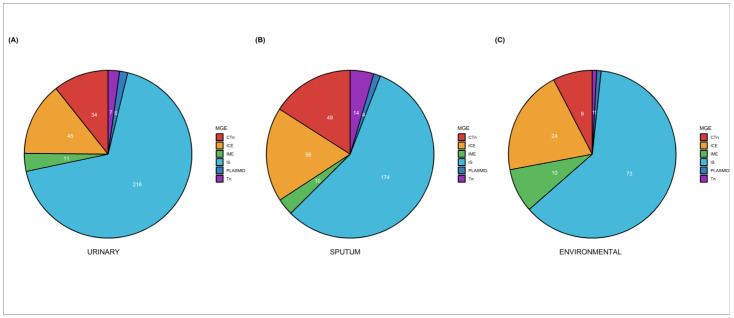
Mobile Genetic Elements (MGEs) are distributed in the 65 genomes of urinary (**A**), sputum (**B**), and environmental (**C**) strains. Each pie graphic represents the distribution of mobile genetic elements in each group of strains, and the numbers within the pie graphic represent the number of MGEs found. Tn (transposons), CTn (composite transposons), IS (insertion sequence), IMEs (integrative mobilizable elements), ICEs (integrative conjugative elements), and plasmids.

**Figure 6 antibiotics-12-00866-f006:**
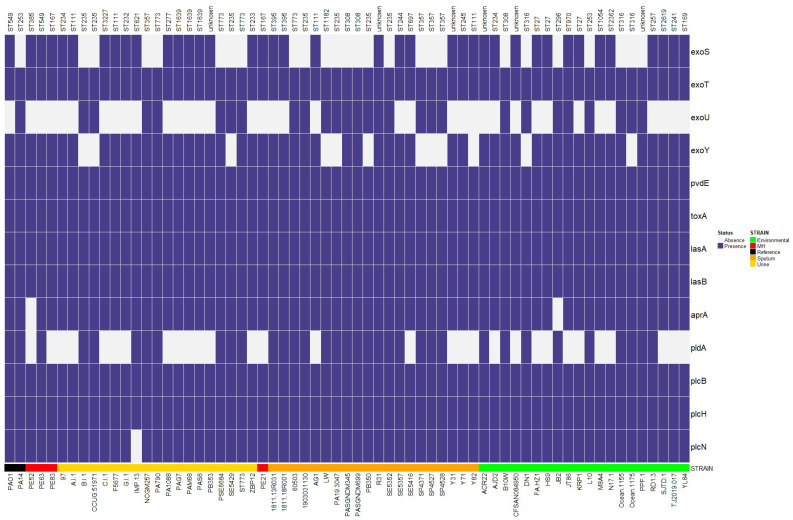
Presence and absence of virulence genes. Heatmap representation of virulence genes associated with urinary and lung infections. Purple represents presence, and gray color represents absence. The genes were identified with VFDB, and the heatmap was constructed with the Bioconductor package in Rstudio. MH means Mexican Hospital.

## Data Availability

The genomes of PE21, PE52, PE63 and PE83 have been deposited in NCBI’s GenBank under accession numbers JARDUV000000000, JARDUU000000000, JARDUW000000000, JARDUX000000000.

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
