# Peer review of "Comparative Genomics of Pseudomonas aeruginosa Strains Isolated from Different Ecological Niches"

_antibiotics, 2023, doi:10.3390/antibiotics12050866_

Round 1
Reviewer 1 Report
The Authors aimed to discuss Comparative genomics of Pseudomonas aeruginosa strains isolated from different ecological niches. The study carries good information that is promising and interesting from a scientific and practical point of view. The authors have deeply described the genetic characteristics particularly STs, virulence factors, and resistance determinants in Pseudomonas aeruginosa strains and graphical representation enlightens the article's information. The manuscript requires minor language and grammar corrections before accepting for publication.
Author Response
Puebla, Pue. México, April, 2023
Dear Reviewers
Microorganism (MDPI)
We appreciate the comments, observations and time invested in reviewing the manuscript.
Title: Comparative genomics of Pseudomonas aeruginosa strains isolated from different ecological niches.
Journal: Microorganism (MDPI)
Submission ID: antibiotics-2301852
We have carefully reviewed all the reviewers’ suggestions. Changes and responses are given point by point, and modifications made in the manuscript paragraphs are marked up with “Track Changes” function.
General corrections
We want to inform you that we have made some corrections to the manuscript. We consider that they are necessary to ensure the accuracy and clarity of our study's findings. The modifications made are detailed in the manuscript, and we have also provided a summary below.
1. In the original manuscript it was mistakenly stated that the most common variant was OXA-395; however, upon further analysis, we have determined that the correct variant is OXA-494 (line 197, Figure S6).
2. For the strains reported with ST167 in the MLST database, we initially identified six strains, however two of them were reads. Therefore, the manuscript was modified to reflect the correct number of strains found and have included their corresponding accession numbers for accuracy (line 356, Table S3).
General Suggestions
I) Please check that all references are relevant to the contents of the manuscript.
References not relevant to the manuscript were removed.
We reviewed the references and removed those not relevant in the manuscript.
II) English language and style are fine/minor spell check required:
The manuscript was corrected and edited for the English language. In addition, a native English speaker made review of the punctuation, spelling, and overall style.
REVIEWER 1
The Authors aimed to discuss Comparative genomics of Pseudomonas aeruginosa strains isolated from different ecological niches. The study carries good information that is promising and interesting from a scientific and practical point of view. The authors have deeply described the genetic characteristics particularly STs, virulence factors, and resistance determinants in Pseudomonas aeruginosa strains and graphical representation enlightens the article's information. The manuscript requires minor language and grammar corrections before accepting for publication.
Your comments and suggestions have helped shape the final version of our paper. The manuscript was corrected and edited for the English language. In addition, a native English speaker made review of the punctuation, spelling, and overall style.
REVIEWER 2
The authors Gómez-Martinez at al. describe genetic features of P. aeruginosa high-risk strains from nosocomial findings in Mexican hospital settings.
The manuscript is well written and easily readable. All figures are presented not only with great graphical representation but on top are detailed.
This manuscript is an going to interest in the scientific community and will be used as a comparative study in how to evaluate high-risk bacterial strains in similar hospital settings internationally.
We have made the corresponding corrections based on your suggestions.
REVIEWER 3
Overall, the referred manuscript is of adequate quality for submission. Given the questions below, I make a few minor suggestions.
1) The study's main question was whether the genome of Pseudomonas aeruginosa showed differences in pathogenic aspects (presence of virulence genes, antimicrobial resistance, mobile genetic elements) in a manner related to the source/origin of isolation of the microorganisms (urinary tract, sputum, and environment). Part of the genomes was obtained by the present study (Mexican clinical isolates), and the rest, from databases.
2) Indeed the work is innovative and contributes to increasing knowledge in this field of study.
3) The number of strains analyzed in the study, and the genomic approach itself are a distinguishing feature, as are the findings on the influence of the environment on adaptability and antibiotic resistance, which differs from clinical isolates.
4) In fact the study presents a suitable control which is a reference strain, highlighted in black in the cladogram.
5) Yes the conclusion is appropriate based on the experimental design and the data presented by the study.
6) Yes, the references are adequate.
7) The figures and tables help in the visualization and understanding of the data presented, and are well organized.
We have made the corresponding corrections based on your suggestions. The manuscript was corrected and edited for the English language. In addition, a native English speaker made review of the punctuation, spelling, and overall style.
REVIEWER 4
Your suggestions were instrumental in guiding our revisions and improving the quality of our research.
I) Line 52: It is important to state some of the factors that make aeruginosa very virulent.
A paragraph was added to the manuscript describing some of the factors that make P. aeruginosa virulent.
Line 56: “The latter is exhibited as a result of the production of a wide variety of virulence factors, including, lipopolysaccharide, flagella, type IV pili, elastase (LasA, LasB), various exotoxins (ExoY, ExoS, ExoT, ExoU), alkaline protease, phospholipases, small molecules (phenazines, rhamnolipids and cyanide) and siderophores (pyoverdine and pyochelin), which are regulated by cell-to-cell signaling systems”.
II) Line 63: Why is it that most of the studies have focused on the behaviour of the strains isolated from eye infections and cystic fibrosis? The reasons for this need to be explained.
In the manuscript is explained why most of the studies have focused on the behavior of the strains isolated from eye infections and cystic fibrosis.
Line 72. “The authors report that these studies were conducted because P. aeruginosa plays an important role in cystic fibrosis infections, whereby, these types of strains have been extensively sequenced”.
III) Line 82: I suggest we have this section being on materials and methods instead of results.
We have carefully considered your feedback regarding the format of our manuscript, and we acknowledge your point about the section on materials and methods. However, the journal Antibiotics requires that the results section be presented before the materials and methods section.
IV) In line 245, it is stated that the environmental genomes showed fewer genetic mobile elements. What explanation can be given for this observation. This should be given in the discussion section.
We have added a possible explanation in the discussion section of the manuscript (line 454), which addresses the question of why the environmental genomes analyzed in our study showed fewer genetic mobile elements than the clinical genomes.
Line 454. “On the other hand, the low number of MGEs in the genomes of environmental strains compared to the genomes of clinical strains, could be related to the fact that they did not have horizontal acquired resistance genes since these are regularly carried by MGEs”.
V) There are sections of the manuscript which need to be revised to make them clearer. I have highlighted these in the reviewed manuscript.
As you suggested, we have carefully reviewed the highlighted sections of the manuscript and made the appropriate corrections. The corrections are marked up with “Track Changes” in the manuscript.
REVIEWER 5
The diagnosis of disease from Pseudomonas aeruginosa has been basically clinical, from symptoms such as some of those described by the author, but the visualization of the catheter-associated urinary tract infections, bloodstream infections, and surgical site infections, please add some sentences about pathophysiology etc.
An explanation of the pathophysiology of infections caused by P. aeruginosa was added.
Line 53. “The pathophysiology of the infections caused by P. aeruginosa involves several mechanisms, including adhesion, invasion, evasion of the immune response, and antibiotic resistance, making these infections difficult to treat”.
Please clarify if there were any carbapenem resistant isolates or genes encoding antibiotic resistance.
Information about carbapenem-resistant isolates in the materials and method section was added (line 578). The resistance genes of each genome are detailed in supplementary materials (Table S2).
Line 578: “Four strains of P. aeruginosa multidrug-resistant and carbapenem resistant recovered from a Mexican Hospital, were included in this study”.
Please add details of MLST analysis, elaborate this part, because methods are most critical pieces of information to make the experiments reproducible!
We have added more details about the MLST analysis in the Methods section (Line 596).
Line 596. “Sequence type (ST) of 65 genomes were identified using the MLST tool from Center for Genomic Epidemiology (https://cge.food.dtu.dk/services/MLST/) [104], which utilizes the MLST allele sequence and profile data from PubMLST.org. The ST is determined by combining the allelic variation of 7 P. aeruginosa housekeeping genes (acsA, aroE, guaA, mutL, nuoD, ppsA, and trpE)”.
The MLST analysis could be extended with hierarchical clustering analysis to show relationships between tested isolates.
We have added a hierarchical clustering analysis of MLST in the manuscript, and the corresponding Figure S5 has been included in the supplementary material.
Line 617. “Seven concatenated sequences of the STs were obtained to the MLST database. MEGA version 11.0.10 was used to construct the MLST dendrogram using the UPGMA method”.
According to the reviewer suggestions, and the changes made, we hope that this manuscript will be considered for publication.
Best regards,
Patricia Lozano Zarain Ph.D.
(Corresponding author)
Centro de Investigaciones en Ciencias Microbiológicas
Instituto de Ciencias. Benemérita Universidad Autónoma de Puebla
Complejo de Ciencias, Edif. IC-11, Ciudad Universitaria Colonia San Manuel, CP 72570, Puebla, Mexico Phone: + 52-222-2295500 ext. 2543
E-mail: plozano_zarain@hotmail.com

Reviewer 2 Report
The authors Gómez-Martinez at al. describe genetic features of P. aeruginosa high-risk strains from nosocomial findings in Mexican hospital settings.
The manuscript is well written and easily readable. All figures are presented not only with great graphical representation but on top are detailed.
This manuscript is an going to interest in the scientific community and will be used as a comparative study in how to evaluate high-risk bacterial strains in similar hospital settings internationally.
Author Response

(The authors gave the same response as above.)

Reviewer 3 Report
Thank you for the contact and the opportunity to act as a reviewer in Antibiotics.
Overall, the referred manuscript is of adequate quality for submission. Given the questions below, I make a few minor suggestions.
1) The study's main question was whether the genome of Pseudomonas aeruginosa showed differences in pathogenic aspects (presence of virulence genes, antimicrobial resistance, mobile genetic elements) in a manner related to the source/origin of isolation of the microorganisms (urinary tract, sputum, and environment). Part of the genomes was obtained by the present study (Mexican clinical isolates), and the rest, from databases.
2) Indeed the work is innovative and contributes to increasing knowledge in this field of study.
3) The number of strains analyzed in the study, and the genomic approach itself are a distinguishing feature, as are the findings on the influence of the environment on adaptability and antibiotic resistance, which differs from clinical isolates.
4) In fact the study presents a suitable control which is a reference strain, highlighted in black in the cladogram.
5) Yes the conclusion is appropriate based on the experimental design and the data presented by the study.
6) Yes, the references are adequate.
7) The figures and tables help in the visualization and understanding of the data presented, and are well organized.
Author Response

(The authors gave the same response as above.)

Reviewer 4 Report
The paper is generally well written but there are some few concerns as follows:-
i) Line 52: It is important to state some of the factors that make P. aeruginosa very virulent
ii) Line 63: Why is it that most of the studies have focused on the behaviour of the strains isolated from eye infections and cystic fibrosis? The reasons for this need to be explained.
iii) Line 82: I suggest we have this section being on materials and methods instead of results.
iv) In line 245, it is stated that the environmental genomes showed fewer genetic mobile elements. What explanation can be given for this observation. This should be given in the discussion section.
v) There are sections of the manuscript which need to be revised to make them clearer. I have highlighted these in the reviewed manuscript.

Author Response

(The authors gave the same response as above.)

Reviewer 5 Report
In the present manuscript, Gómez-Martínez et al. Comparative genomics of Pseudomonas aeruginosa strains isolated from different ecological niches. The basic idea is quite interesting.
Although, the scope and originality of the study is better presented throughout the manuscript, but there are minor corrections needed in the manuscript.
The diagnosis of disease from Pseudomonas aeruginosa has been basically clinical, from symptoms such as some of those described by the author, but the visualization of the catheter-associated urinary tract infections, bloodstream infections, and surgical site infections, please add some sentences about pathophysiology etc.
Please clarify if there were any carbapenem resistant isolates or genes encoding antibiotic resistance.
Please add details of MLST analysis, elaborate this part, because methods are most critical pieces of information to make the experiments reproducible!
The MLST analysis could be extended with hierarchical clustering analysis to show relationships between tested isolates.
Author Response

(The authors gave the same response as above.)

Round 2
Reviewer 5 Report
I appreciate the authors, they improved the manuscript. Now I recommend this revised manuscript for publication.